# Relationship of Serum Uric Acid with Kidney Function Decline Mediated by Systemic Arterial Stiffness: A Retrospective Cohort Study in Japan

**DOI:** 10.3390/diagnostics14020195

**Published:** 2024-01-16

**Authors:** Daiji Nagayama, Yasuhiro Watanabe, Kentaro Fujishiro, Kenji Suzuki, Kohji Shirai, Atsuhito Saiki

**Affiliations:** 1Department of Internal Medicine, Nagayama Clinic, Oyama-City 323-0032, Tochigi, Japan; 2Center of Diabetes, Endocrinology and Metabolism, Toho University, Sakura Medical Center, Sakura-City 285-0841, Chiba, Japan; 601055wy@med.toho-u.ac.jp (Y.W.); atsuhito156@sakura.med.toho-u.ac.jp (A.S.); 3Japan Health Promotion Foundation, Shibuya-ku 150-0013, Tokyo, Japan; kennyfuji@icloud.com (K.F.); kenjiji054@gmail.com (K.S.); 4Department of Internal Medicine, Mihama Hospital, Chiba-City 261-0013, Chiba, Japan; kshirai@kb3.so-net.ne.jp

**Keywords:** serum uric acid, kidney function decline, arterial stiffness, cardio-ankle vascular index

## Abstract

Hyperuricemia is associated with kidney function decline (KFD), although whether hyperuricemia directly causes nephrotoxicity or is indirectly mediated by systemic arterial stiffening remains unclear. We examined the detailed relationship of serum uric acid (SUA) with KFD and potential mediation by arterial stiffness. Study population was 27,648 urban residents with an estimated glomerular filtration rate (eGFR) of ≥60 mL/min/1.73 m^2^ at baseline, and they participated in a median of three consecutive annual health examinations. Arterial stiffness was assessed using cardio-ankle vascular index (CAVI). KFD was defined as a decrease in eGFR to below 60. Multivariate analysis showed an association between baseline SUA and CAVI independent of eGFR. During the study period, 6.6% of participants developed KFD. Stratified analysis revealed a linear relationship between the contribution of CAVI or SUA and KFD. ROC analysis determined a cutoff CAVI of 8.0 (males) or 7.9 (females) and a cutoff SUA of 6.3 (males) or 4.5 mg/dL (females) for predicting KFD. The linkage between SUA and CAVI was associated with a greater increase in the hazard ratio for KFD with an increase in SUA. CAVI showed the mediating effect on the relationship of SUA with KFD after an adjustment for confounders. SUA was associated positively with CAVI-mediated KFD. Further studies should verify whether intensive SUA-lowering treatment prevents KFD via improving vascular function.

## 1. Introduction

Chronic kidney disease (CKD) and end-stage kidney disease are serious global health issues, leading to reduced quality of life and premature mortality. According to a recent review, CKD is the 16th leading cause of years of life lost worldwide, underscoring the importance of appropriate screening, diagnosis, and management by primary care clinicians to reduce the burden of CKD [1]. Elevated serum uric acid (SUA) levels seen in CKD may lead to renal tubular injury, endothelial dysfunction, oxidative stress, and inflammation. Furthermore, hyperuricemia can also be a risk factor that precedes the development of CKD and can be considered an upstream pathophysiology that accelerates kidney injury [2]. In other words, kidney function decline (KFD) induces hyperuricemia by decreasing uric acid excretion, and hyperuricemia further accelerates kidney injury, forming a vicious cycle [3]. On the other hand, reports also show that the disorders of uric acid metabolism may promote systemic arteriosclerosis independent of KFD [4,5]. Taken together, the associations of uric acid with KFD and systemic arteriosclerosis involve a complex causal relationship.

Systemic arterial stiffness reflects vascular function, and arterial stiffness parameter is useful for the evaluation and the treatment of cardio-metabolic disorders [6]. Pulse wave velocity (PWV) is the most widely used arterial stiffness parameter and has been confirmed to be associated with cardiovascular events [7]. Cardio-ankle vascular index (CAVI), developed after PWV, is also an arterial stiffness parameter that includes the entire arterial tree from the aortic valve to the tibial artery [8]. Unlike PWV, CAVI has been theoretically and clinically proven to be independent of blood pressure (BP) at the time of measurement. Reports have shown that vascular toxicity caused by various cardio-metabolic risk factors can be extracted by CAVI, and that appropriate therapeutic interventions can improve CAVI. Furthermore, several cross-sectional studies have reported that CAVI is associated with kidney function indicators including estimated glomerular filtration rate (eGFR), serum cystatin C [9], and renal resistive index [10]. Additionally, CAVI has been reported to predict not only cardio-vascular events [11,12] but also KFD more effectively than PWV in the general Japanese population [13]. Other than the associations with kidney function indicators, an increased CAVI was also independently and linearly associated with an increased serum SUA in our previous cross-sectional study [14]. However, it remains unclear whether the nephrotoxicity of hyperuricemia is a direct effect or is indirectly mediated by systemic arterial stiffening.

With the above background, the present retrospective cohort study aimed to elucidate in detail how CAVI and SUA are associated with KFD, and whether CAVI has a mediating effect on the relationship between SUA and KFD.

## 2. Materials and Methods

### 2.1. Individuals and Design

This research was a retrospective cohort study using data from the cardiovascular disease and cancer screening program organized by the Japan Health Promotion Foundation. The study population was 34,662 Japanese residents of major cities in Japan who had undergone 2 to 8 consecutive annual health checkups between 2010 and 2018. Of 34,662 individuals evaluated for eligibility, those with insufficient data (*N* = 5026) and those with an eGFR of less than 60 mL/min/1.73 m^2^ at the first examination (*N* = 1952) were excluded. Finally, 27,684 individuals without kidney function impairment at baseline were included in the study.

### 2.2. Data Collection

All parameters were assessed in a standardized manner. Height and weight (BW) were measured, and body mass index (BMI) was calculated as BW (kg)/height (m) squared. Blood pressure (BP) was measured from an upper arm cuff, in the sitting position after 5 min of rest. Hypertension was diagnosed by either systolic blood pressure (SBP) of ≥140 mmHg, diastolic blood pressure of ≥90 mmHg, or treatment with BP-lowering drugs.

Blood sampling was conducted from an anterior upper extremity vein in the morning after a 12 h fast to measure fasting plasma glucose (FPG, mg/dL), SUA (mg/dL), triglycerides (TG, mg/dL), and high-density lipoprotein cholesterol (HDL-C, mg/dL). SUA was measured using the uricase–peroxidase method. Low-density lipoprotein cholesterol (LDL-C) (mg/dL) was calculated using Friedewald’s formula: LDL-C = (TC) − (HDL-C) − (TG/5). This formula is invalid for patients with TG ≥ 400 mg/dL. Therefore, individuals with TG ≥ 400 mg/dL (*N* = 295, 1.07%) were excluded only from the LDL-C analysis. To correct the deviation from normality, only TG was transformed to a natural log scale; i.e., ln(TG). Diabetes mellitus was diagnosed when FPG was ≥126 mg/dL or when the participant was on antidiabetic drugs. Dyslipidemia was defined as LDL-C ≥ 140 mg/dL, HDL-C < 40 mg/dL, and/or TG ≥ 150 mg/dL, or treatment with lipid-lowering drugs.

eGFR was calculated using the following formula from the Japanese Society of Nephrology [15]:eGFR (mL/min/1.73 m^2^) = 194 × creatinine^−1.094^ × age^−0.287^ (× 0.739 if female).

KFD was defined as eGFR below 60 mL/min/1.73 m^2^, corresponding to GFR category 3a or worse [16]. Since all the individuals had an eGFR of 60 mL/min/1.73 m^2^ or higher at the initial health examination, a decrease in eGFR to below 60 mL/min/1.73 m^2^ at any subsequent annual examination during the study period was defined as the development of KFD. The rate of change in eGFR per year (% ΔeGFR/y) was calculated for the period up to follow-up.

Spot urine samples were collected and used for urinalysis with the dipstick method. Urinalysis results were recorded as (−), (±), (1+), (2+), and (3+). Proteinuria was defined as urinary protein (1+) or above, which corresponds to a urine protein level of 30 mg/dL or higher.

The prevalence of metabolic disorders, current smoking, and habitual alcohol consumption were determined using a questionnaire. Habitual alcohol consumption was defined as daily drinking.

### 2.3. Measurement of Arterial Stiffness Parameters and Blood Pressure

CAVI was estimated using the VaSera VS-1500 device (Fukuda Denshi Co, Ltd., Tokyo, Japan) according to the manufacturer’s instructions. CAVI values were automatically calculated using the following formula [8]:CAVI = a{2ρ × ln(Ps/Pd)/ΔP × PWV^2^} + b,
where Ps means SBP; Pd means DBP; ΔP is Ps − Pd; ρ is blood density; PWV denotes cardio-ankle PWV; and a and b are constants.

Cuffs were placed on the arms and legs, and a heart sound microphone was attached with a double-sided tape to the sternum at the second intercostal space. Participants remained still and silent for 5 min of measurement.

### 2.4. Definition of Linkage between SUA and CAVI

In the present study, we examined whether the linkage between SUA and CAVI alters the nephrotoxicity of SUA. The participants were divided into two groups according to the presence or absence of the linkage. First, the cutoff values of SUA and CAVI for predicting KFD were calculated by receiver operating characteristic (ROC) curve analysis. Hyperuricemia or high CAVI was defined as SUA or CAVI at or above the cutoff value. Next, the linkage between SUA and CAVI was defined as either “hyperuricemia (+) and high CAVI (+)” or “hyperuricemia (−) and high CAVI (−)”. On the other hand, “hyperuricemia (+) and high CAVI (−)” or “hyperuricemia (−) and high CAVI (+)” indicated no linkage.

### 2.5. Statistical Analysis

All data are expressed as medians (interquartile ranges) or percentages. Mann–Whitney U test or Fisher’s exact test was performed to examine differences in baseline characteristics between individuals with and those without KFD. The relationship between CAVI and clinical variables was analyzed using univariate and multivariate linear regression analyses. Cox-proportional hazards analysis was also performed to identify the contribution of the variables to KFD, and the result is expressed as hazard ratio with 95% confidence interval (CI). Wald test was conducted to assess the probability of the null hypothesis of no difference in all groups stratified by CAVI or SUA. Using ROC curve analysis combined with Youden’s J index, the areas under the ROC curves with 95% CI were calculated, and the best discriminating level of CAVI or SUA for predicting KFD was determined. Mediation analysis was carried out using PROCESS (version 4.0) in SPSS [17]. The total effect must be significant to ensure the presence of mediation. Partial mediation exists when both indirect and direct effects are significant. The mediation rate (%) indicates the contribution of the mediation to the total effect and is calculated using the following formula: indirect effect/total effect × 100. In all comparisons, two-sided *p* values less than 0.05 were considered statistically significant. The SPSS software (version 27.0.1, Chicago, IL, USA) was used for all statistical analyses.

## 3. Results

### 3.1. Clinical Characteristics of Participants with or without KFD

Of the 27,648 participants (median age: 46 years; male: 44.4%), 1820 (6.6%) developed KFD during the study period. Table 1 compares the baseline clinical characteristics of participants who did and those who did not develop KFD. The group with KFD had significantly higher male ratio, age, BMI, CAVI, BP, FPG, TC, LDL-C, TG, creatinine, SUA, the frequency of proteinuria, and the frequencies of current treatments for hypertension, diabetes, dyslipidemia, and gout/hyperuricemia as well as significantly lower HDL-C, eGFR, and % ΔeGFR/y. In addition, the frequency of current smoking tended to be lower in the group with KFD.

Based on these findings, the following representative confounders were used in the subsequent analyses to calculate the contribution of CAVI or SUA to KFD: age, sex, BMI, SBP, FPG, proteinuria, and the frequency of current smoking.

### 3.2. Correlation of Baseline CAVI with Clinical Variables

As shown in Table 2, simple univariate analysis revealed that CAVI correlated positively with age, male sex, BP, FPG, LDL-C, ln(TG), and SUA and negatively with current smoking, habitual alcohol consumption, height, BMI, and eGFR. In a multivariate analysis using these variables, age, male sex, current smoking, habitual alcohol drinking consumption, BMI, FPG, ln(TG), eGFR, and SUA were extracted as significant independent factors associated with CAVI. Height, DBP, and HDL were not recruited for the multivariate model due to their intra-class correlations with BMI, SBP, and lnTG, respectively. Proteinuria did not show a significant univariate correlation with CAVI and was therefore not recruited as well.

### 3.3. Relationship of Adjusted Hazard Ratio for KFD with Stratified CAVI and SUA

The relationship of the contribution of CAVI or SUA to KFD was examined by the stratified analysis of hazard ratio obtained from Cox-proportional hazards analysis. Figure 1 shows the relationship of adjusted hazard ratio for KFD with the strata of baseline CAVI or SUA. 

According to Wald test for trends, the increasing CAVI or SUA strata was significantly associated with an increase in the hazard ratio for KFD. When CAVI of 6.9 was adopted as a reference, CAVI of 7 and above showed a significant increase in the hazard ratio. On the other hand, when SUA of 2.9 mg/dL was adopted as a reference, SUA of 5 mg/dL and above showed a significant increase in the hazard ratio. When similar analyses were conducted separately for men and women, the distribution of SUAs had a large gender difference, which widened the 95% CI and made it difficult to obtain meaningful results.

### 3.4. Cutoff Values of CAVI and SUA for Predicting KFD Obtained from ROC Curve Analysis

The diagnostic accuracy of CAVI or SUA for KFD identified using ROC curve analysis is shown in Table 3. In this analysis, both CAVI and SUA showed a significant predictive accuracy for KFD in both sexes. The optimal cutoff values were CAVI 8.0 (males) or 7.9 (females) and SUA 6.3 (males) or 4.5 (females) mg/dL for predicting KFD.

### 3.5. Effects of the Linkage between SUA and CAVI on the Relationship of SUA with KFD

Based on the results of the ROC curve analyses, hyperuricemia was defined by SUA of ≥4.5 mg/dL in females or ≥6.3 mg/dL in males and a high CAVI of ≥ 8.0 in males or ≥7.9 in females. Accordingly, the linkage between SUA and CAVI was defined as either “hyperuricemia (+) and high CAVI (+)” or “hyperuricemia (−) and high CAVI (−)”, and no linkage was “hyperuricemia (+) and high CAVI (−)” or “hyperuricemia (−) and high CAVI (+)”.

Figure 2A shows the incidence of KFD versus SUA tiers when individuals were divided based on the linkage between SUA and CAVI and no linkage. Individuals with the linkage (gray bar) showed an increasing incidence of KFD with an increase in SUA, whereas those with no linkage (white bar) showed no remarkable differences.

Next, the contribution of one standard deviation increase in SUA to the risk of developing KFD according to the presence or absence of the linkage was examined, as shown in Figure 2B. In this analysis, individuals with the linkage (gray bar) showed higher hazard ratios for KFD than those with no linkage (white bar), suggesting that the linkage between SUA and CAVI may accelerate the nephrotoxicity of uric acid.

### 3.6. Mediation Analysis of CAVI as Potential Mediator of the Association between SUA and KFD

Finally, as shown in Figure 3, we examined whether the relationship between SUA and KFD was mediated by CAVI. In this analysis, SUA and CAVI independently contributed to the development of KFD, respectively. Furthermore, a partial mediation effect of CAVI on KFD was observed, and the effect remained significant even after adjustment for confounding factors (mediation rate = 7.3%).

## 4. Discussion

In the present retrospective cohort study that analyzed individuals with normal kidney function at the initial health examination, kidney function declined in 6.6% of all the participants during the study period. At baseline, SUA was positively associated with CAVI independent of eGFR. SUA and CAVI were independently and positively associated with the risk of KFD. The linkage between SUA and CAVI augmented the risk of KFD with an increase in SUA. Furthermore, a positive relationship of SUA with CAVI-mediated KFD was also observed. The novelty of this study is that SUA was associated positively with KFD via systemic arterial stiffening in individuals participating in consecutive annual health examinations. 

The mechanisms by which hyperuricemia impairs kidney function can be divided into two pathways: (1) a direct pathway termed gouty nephropathy, and (2) an indirect pathway mediated by systemic arteriosclerosis. The pathophysiology of the former, i.e., gouty kidney, is chronic inflammation caused by sodium urate crystals deposited in the renal tubules and interstitium [18,19]. With regard to the indirect pathway, i.e., the pathophysiology mediated by systemic arteriosclerosis, several candidate factors have been identified, such as the activation of the renin–angiotensin–aldosterone system, the inhibition of nitric oxide synthesis, and an increased oxidative stress [20,21]. Given the susceptibility of the kidneys to hemodynamic changes in the presence of large volumes of blood flow, it is reasonable to hypothesize that systemic arterial stiffening with increased pulsatile flow impairs kidney function [22]. In the present study, the participants were divided into two groups according to the presence or absence of the linkage between SUA and CAVI. An increase in SUA was associated with the risk of KFD, especially in individuals with the linkage, suggesting hyperuricemia-induced nephrotoxicity via systemic arterial stiffening. Since our data reveal that CAVI reflects the indirect pathway of hyperuricemia-induced nephrotoxicity, individuals with hyperuricemia and an increased CAVI would require intensive SUA-lowering therapy as early as possible to prevent KFD. Furthermore, CAVI may be a useful indicator to evaluate the therapeutic effect. Further verification is required in the future to clarify the pathophysiology of the linkage between SUA and CAVI.

The upstream pathophysiology in the uric acid synthesis pathway may also accelerate systemic arteriosclerosis. Xanthine oxidase that catalyzes the oxidation of xanthine and hypoxanthine to uric acid is known to induce vascular injury by generating superoxide anions [23,24]. Accordingly, it remains controversial whether SUA is a pathogenic mediator or an atherogenic marker of systemic arterial stiffening. Kario et al. [25] reported that CAVI reduction was not observed after 24 weeks of treatment with either of two xanthine oxidase inhibitors, topiroxostat or febuxostat, in hypertensive patients with hyperuricemia in the BEYOND UA study. However, a sub-analysis of the above study found that topiroxostat reduced CAVI in a sub-population with liver dysfunction [26]. On the other hand, dotinurad, a urate reabsorption inhibitor, also reduced CAVI despite not inhibiting xanthine oxidase [27]. Considering these previous reports indicating that the inhibition of xanthine oxidase is not necessarily required to improve CAVI, it seems less relevant to consider SUA as just an atherogenic marker. If SUA per se is a pathogenic mediator, then lowering the SUA level would be useful to prevent KFD, irrespective of whether the SUA-lowering mechanism is the inhibition of xanthine oxidase or the acceleration of uric acid excretion. Future study is needed to verify whether the degree of CAVI improvement due to SUA-lowering therapy relates to the prevention of KFD.

In this study, the cutoff value of SUA for KFD showed a sex difference, unlike the cutoff value of CAVI. We have already reported a cross-sectional study showing a similar sex difference in the SUA range for an increase in CAVI [14]. It remains unclear why the magnitude of the association between SUA and KFD differs between males and females. In terms of CKD prevention, primary care clinicians should not underestimate hyperuricemia, especially in women.

A J-shaped relationship between the SUA level and the risk of kidney injury has been reported recently. Specifically, several longitudinal studies have shown that even low serum SUA levels are associated with a decrease in eGFR [28,29,30]. It has been known since the early 1980s that uric acid has a strong antioxidant effect [31,32]. Accordingly, individuals with hypouricemia may be susceptible to the organic free radicals constantly generated by the activities of daily living. Therefore, hypouricemia increases oxidative stress and induces post-exercise renal artery spasm, resulting in the loss of kidney function [33,34]. However, in the present study, in which KFD was defined as the decrease in eGFR to below 60 mL/min/1.73 m^2^ during the study period, the group with the lowest SUA showed the lowest risk of KFD (Figure 1B). This finding is incompatible with the aforementioned nephrotoxicity induced by hypouricemia. As a hypothesis to explain this discrepancy, we postulate that uric acid excretion into urine decreases with progressive kidney dysfunction [2]. In other words, the nephrotoxicity of hypouricemia may eventually disappear as SUA increases with decreasing eGFR.

The present study has several limitations. First, the cutoff values of CAVI and SUA, which were independently associated with KFD in this study, may only be applied to the middle-aged Japanese population. Next, the dipstick method was used to determine urinary protein, and this test does not provide precise results. Finally, using a cohort that participates in annual physical examinations has a potential selection bias. Individuals who regularly participate in health screenings may be more health conscious and have a healthier lifestyle than those who do not participate. Therefore, caution should be exercised when generalizing the findings of this study to the general population.

## 5. Conclusions

SUA was positively associated with CAVI-mediated KFD. Further studies should verify whether intensive SUA-lowering treatment prevents KFD via improving vascular function.

## Figures and Tables

**Figure 1 diagnostics-14-00195-f001:**
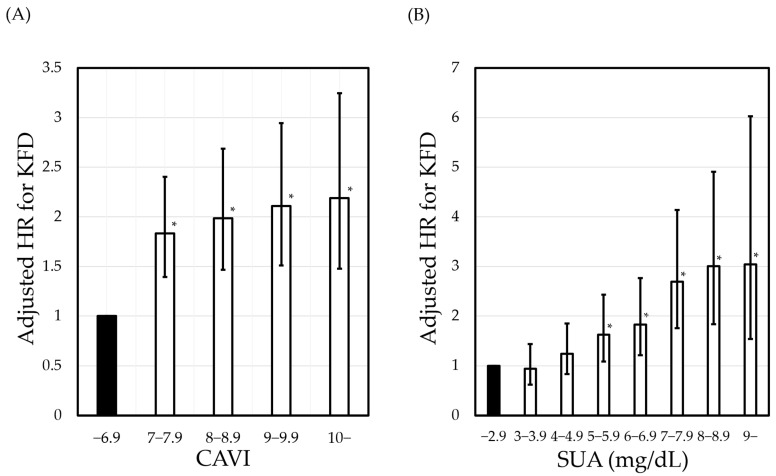
Relationship of adjusted hazard ratio for KFD with stratified (**A**) CAVI and (**B**) SUA. Hazard ratios with 95% confidence interval are shown. * Significant increase in hazard ratio versus the lowest stratum as the control group. *p* values for trend were estimated using Wald test. Hazard ratios were adjusted for age, sex, BMI, SBP, FPG, LDL-C, HDL-C, ln(TG), current smoking, proteinuria, and the treatment of gout and/or hyperuricemia. KFD was defined as a decrease in eGFR to below 60 mL/min/1.73 m^2^ during the study period. HR, hazard ratio; KFD, kidney function decline; CAVI, cardio-ankle vascular index; SUA, serum uric acid; BMI, body mass index; SBP, systolic blood pressure; FPG, fasting plasma glucose; LDL-C, low-density lipoprotein cholesterol; HDL-C, high-density lipoprotein cholesterol; TG, triglycerides; and eGFR, estimated glomerular filtration rate.

**Figure 2 diagnostics-14-00195-f002:**
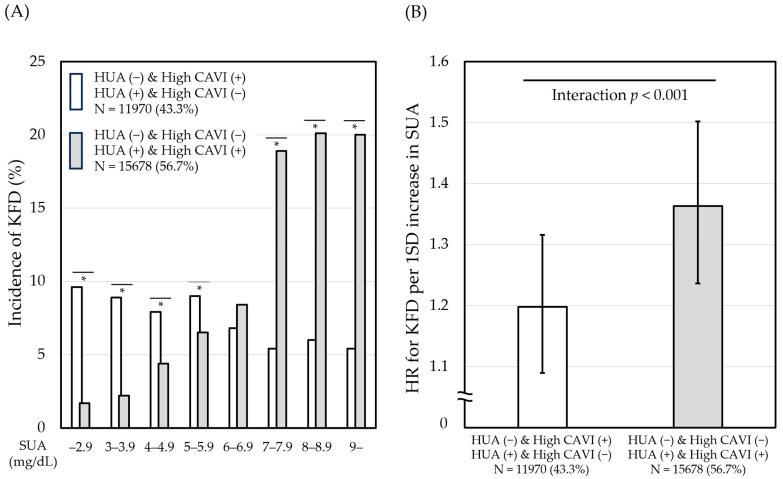
Effects of the linkage between SUA and CAVI on the relationship of SUA with kidney function decline. (**A**) Association of SUA tiers with the incidence of KFD (%). * *p* < 0.001, Fischer’s exact test. (**B**) Hazard ratio for KFD per one SD increase in SUA. Hazard ratios (95% confidence interval) were adjusted for age, sex, BMI, SBP, FPG, LDL-C, HDL-C, ln(TG), current smoking, proteinuria, and the treatment of gout and/or hyperuricemia in Cox-proportional hazard analysis. KFD was defined as a decrease in eGFR to below 60 mL/min/1.73 m^2^ during the study period. Hyperuricemia was defined as SUA ≥ 4.5 mg/dL in females or ≥6.3 mg/dL in males. High CAVI was defined as CAVI ≥ 8.0 in males or ≥7.9 in females. KFD, kidney function decline; HR, hazard ratio; CAVI, cardio-ankle vascular index; SD, standard deviation; SUA, serum uric acid; BMI, body mass index; SBP, systolic blood pressure; FPG, fasting plasma glucose; LDL-C, low-density lipoprotein cholesterol; HDL-C, high-density lipoprotein cholesterol; TG, triglycerides; and eGFR, estimated glomerular filtration rate.

**Figure 3 diagnostics-14-00195-f003:**
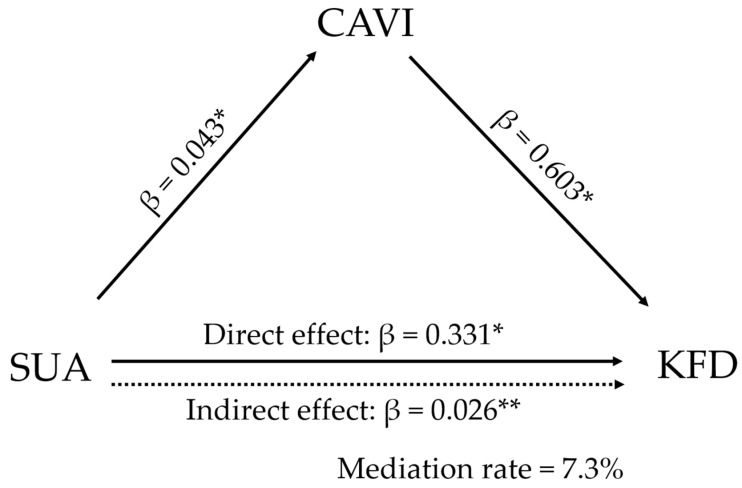
Mediation analysis of CAVI as a potential mediator for the association between SUA and KFD. KFD was defined as a decrease in eGFR to below 60 mL/min/1.73 m^2^ during the study period. Analysis was adjusted for confounding factors of older age (≥65 years), sex, BMI ≥ 25 kg/m^2^, hypertension, diabetes mellitus, dyslipidemia, the treatment of gout/hyperuricemia, current smoking, and proteinuria. KFD, kidney function decline; CAVI, cardio-ankle vascular index; SUA, serum uric acid; and BMI, body mass index. * *p* < 0.001, ** *p* < 0.05. β, standardized β coefficient.

**Table 1 diagnostics-14-00195-t001:** Clinical characteristics of participants with or without KFD.

Variable	Individuals without KFD (*N* = 25,828)	Individuals with KFD (*N* = 1820)	*p* Value
Male gender (%)	44.0	46.9	0.018
Age (years)	44 (36–56)	61 (52–68)	<0.001
Height (meter)	1.62 (1.56–1.70)	1.61 (1.55–1.68)	<0.001
BMI (kg/m^2^)	21.9 (19.9–24.2)	22.9 (20.8–24.9)	<0.001
SBP (mmHg)	116 (107–127)	126 (113–138)	<0.001
DBP (mmHg)	72 (65–80)	78 (70–86)	<0.001
CAVI	7.4 (6.9–8.2)	8.4 (7.7–9.1)	<0.001
FPG (mg/dL)	85 (80–91)	89 (84–97)	<0.001
TC (mg/dL)	208 (185–235)	217 (195–241)	<0.001
LDL-C (mg/dL)	121 (100–144)	130 (109–150)	<0.001
HDL-C (mg/dL)	67 (56–81)	63 (53–77)	<0.001
TG (mg/dL)	78 (55–117)	96 (69–139)	<0.001
ln(TG)	4.36 (4.01–4.76)	4.56 (4.23–4.93)	<0.001
Creatinine (mg/dL)	0.68 (0.59–0.81)	0.77 (0.69–0.92)	<0.001
eGFR (mL/min/1.73 m^2^)	82.6 (74.6–92.0)	65.6 (62.6–69.6)	<0.001
% ΔeGFR/y	−1.34 (−3.33–0.69)	−3.26 (−5.21–−1.15)	<0.001
SUA (mg/dL)	4.9 (4.0, 5.9)	5.3 (4.4, 6.3)	<0.001
SBP ≥ 140 and/or DBP ≥ 90 mmHg (%)	12.7	27.5	<0.001
FPG ≥ 126 mg/dL (%)	1.3	3.1	<0.001
LDL-C ≥ 140, HDL-C < 40 and/or TG ≥ 150 mg/dL (%)	37.8	49.1	<0.001
Proteinuria (%)	5.0	7.4	<0.001
Current smoking (%)	35.1	32.9	0.061
Habitual alcohol drinking (%)	34.2	33.1	0.331
Receiving treatment for			
Hypertension (%)	7.1	22.8	<0.001
Diabetes mellitus (%)	3.9	6.8	<0.001
Dyslipidemia (%)	6.0	15.1	<0.001
Gout/Hyperuricemia (%)	0.9	3.2	<0.001

Data are presented as medians (interquartile ranges) or percentages. Comparison of two groups was performed using Mann–Whitney U test for continuous variables and Fisher’s exact test for dichotomous variables. KFD was defined as a decrease in eGFR to below 60 mL/min/1.73 m^2^ during the study period. KFD, kidney function decline; BMI, body mass index; SBP, systolic blood pressure; DBP, diastolic blood pressure; CAVI, cardio-ankle vascular index; FPG, fasting plasma glucose; TC, total cholesterol; LDL-C, low-density lipoprotein cholesterol; HDL-C, high-density lipoprotein cholesterol; TG, triglycerides; eGFR, estimated glomerular filtration rate; and SUA, serum uric acid.

**Table 2 diagnostics-14-00195-t002:** Correlation of baseline CAVI with clinical variables.

CAVI vs.	Univariate		Multivariate *	
	ρ	*p* Value	Standardized β	*p* Value
Age (years)	0.796	<0.001	0.766	<0.001
Sex (male, 1; female, 0)	0.020	0.001	0.063	<0.001
Current smoking	−0.072	<0.001	−0.013	0.002
Habitual alcohol consumption	−0.049	<0.001	0.009	0.028
Height (meter)	−0.172	<0.001		
BMI (kg/m^2^)	−0.055	<0.001	−0.160	<0.001
SBP (mmHg)	0.402	<0.001	0.116	<0.001
DBP (mmHg)	0.417	<0.001		
FPG (mg/dL)	0.394	<0.001	0.052	<0.001
LDL-C (mg/dL)	0.246	<0.001	<0.001	0.965
HDL-C (mg/dL)	0.008	0.168		
TG (mg/dL)	0.230	<0.001		
ln(TG)	0.230	<0.001	0.037	<0.001
eGFR (mL/min/1.73 m^2^)	−0.386	<0.001	−0.018	<0.001
SUA (mg/dL)	0.067	<0.001	0.039	<0.001
Proteinuria	−0.003	0.572		

Univariate analysis was performed using Spearman’s rank correlation analysis. * Model: R^2^ = 0.718, *p* < 0.001. CAVI, cardio-ankle vascular index; BMI, body mass index; SBP, systolic blood pressure; DBP, diastolic blood pressure; FPG, fasting plasma glucose; LDL-C, low-density lipoprotein cholesterol; HDL-C, high-density lipoprotein cholesterol; TG, triglycerides; eGFR, estimated glomerular filtration rate; SUA, serum uric acid.

**Table 3 diagnostics-14-00195-t003:** Cutoff values of CAVI and SUA for predicting KFD obtained from receiver operating characteristic curve analysis.

Parameter	Sex	Cutoff	Sensitivity	Specificity	AUC (95% CI)	*p* Value
CAVI	Male	8.0	0.708	0.687	0.757 (0.741–0.772)	<0.001
	Female	7.9	0.669	0.663	0.726 (0.711–0.742)	<0.001
SUA	Male	6.3	0.500	0.614	0.564 (0.544–0.584)	<0.001
	Female	4.5	0.572	0.594	0.614 (0.595–0.632)	<0.001

Youden’s J Index was used in conjunction with receiver operating characteristic curve analysis to select the optimal cutoff values of CAVI and SUA for predicting KFD, defined as a decrease in eGFR to below 60 mL/min/1.73 m^2^ during the study period. CAVI, cardio-ankle vascular index; KFD, kidney function decline; SUA, serum uric acid; AUC, area under the receiver operating characteristic curve; 95% CI, 95% confidence interval; and eGFR, estimated glomerular filtration rate.

## Data Availability

The data that support the findings of this study are not publicly available because they contain information that could compromise the privacy of research participants. Further enquiries may be directed to the corresponding author.

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
