# Peer review of "Relationship of Serum Uric Acid with Kidney Function Decline Mediated by Systemic Arterial Stiffness: A Retrospective Cohort Study in Japan"

_diagnostics, 2024, doi:10.3390/diagnostics14020195_

Round 1

Reviewer 1 Report

Comments and Suggestions for Authors

1.       Line 134: Does "multivariate regression" in this context refer to "multiple linear regression"?

2.       Line 148-162: Kindly omit this section from the template.

3.       Table 1: Are the variables following SUA (mg/dL) up to before proteinuria (%) presented in percentage? Please include the unit.

4.       Table 2: Could you clarify the reason for the absence of results from multivariate analysis for height, DBP, HDL, TG, and proteinuria?

5.       In Table 3, the authors established the cut-off value for CAVI and SUA by considering gender distinctions. Could you kindly clarify why a similar approach was not employed to determine the HR in Figure 1?

Author Response

Reply to reviewer's comments.

We are grateful to reviewer 1 for the critical comment and useful suggestion that have helped us to improve our paper considerably. As indicated in the response that follow, we have taken the comments and suggestion into account in the revised version of our paper.

Comment 1.

Line 134: Does "multivariate regression" in this context refer to "multiple linear regression"?

Response:

That is right. We thank reviewer 1 for the appropriate remark; 'linear' has been added to the relevant text.

Comment 2.

Line 148-162: Kindly omit this section from the template.

Response:

We thank reviewer 1 for appropriately pointing out our careless mistake; the relevant text has been deleted.

Comment 3.

Table 1: Are the variables following SUA (mg/dL) up to before proteinuria (%) presented in percentage? Please include the unit.

Response:

We thank reviewer 1 for appropriately pointing out our careless mistake; (%) has been added to the relevant part of Table 1.

Comment 4.

Table 2: Could you clarify the reason for the absence of results from multivariate analysis for height, DBP, HDL, TG, and proteinuria?

Response:

We appreciate reviewer 1 for the pertinent comment which complement our lack of explanation. Usually, only factors showing significant differences in a univariate correlation analysis can be adopted in the subsequent multiple regression analysis. However, in such cases it is undesirable to adopt factors showing an intra-class correlation at the same time. As an explanation, we have added the following text to the relevant result section;

“Height, DBP and HDL were not recruited for the multivariate model due to their intra-class correlations with BMI, SBP and lnTG, respectively. Proteinuria did not show a significant univariate correlation with CAVI and was therefore not recruited as well.”

Comment 5.

In Table 3, the authors established the cut-off value for CAVI and SUA by considering gender distinctions. Could you kindly clarify why a similar approach was not employed to determine the HR in Figure 1?

Response:

The comment is a very important point. We initially analyzed separately for men and women. For CAVI, the result was similar to that in Fig. 1(A), even if we split the analysis by sex. For SUA, on the other hand, a similar analysis separately for men and women did not yield significant hazard ratios for many SUA ranges, due to larger 95% CIs. This could be due to the fact that the distribution of SUA has large gender difference and therefore did not ensure a sufficient number of participants for the analysis. In other words, when analyzed separately for men and women, it was no longer possible to clearly demonstrate that SUA independently contributes to KFD. We have therefore created Figure 1 with the sole aim of presenting the hazard ratios for KFD for each CAVI and SUA strata, without taking gender differences into account.

In response to the comment, we have added the following text to the relevant result section; “When similar analyses were conducted separately for men and women, the distribution of SUAs had a large gender difference, which widened the 95% CI and made it difficult to obtain meaningful results.”

Reviewer 2 Report

Comments and Suggestions for Authors

The topic of the article is very interesting. Although there are many articles evaluating the correlation between SUA and CAVI (including one published by some of the authors of the current one) the merit of this article is that is evaluates the impact of these parameters on the kidney function decline. The number of cases evaluated is impresive. 

The formulation “improving arterial stiffness” is not the most adequate - please rephrase.

At the end of Statistical analysis section there is some text that is not part of the article. It seems that is part of the template used for article submission - please remove

Comments on the Quality of English Language

Minor correction are required

Author Response

Reply to reviewer's comments.

We are grateful to reviewer 2 for the critical comment and useful suggestion that have helped us to improve our paper considerably. As indicated in the response that follow, we have taken the comments and suggestion into account in the revised version of our paper.

Comment 1.

The formulation “improving arterial stiffness” is not the most adequate - please rephrase.

Response:

We appreciate reviewer 2 for pointing out the inappropriate expression. Changed "improve arterial stiffness" to "improve vascular function" where applicable.

Comment 2.

At the end of Statistical analysis section there is some text that is not part of the article. It seems that is part of the template used for article submission - please remove.

Response:

We thank reviewer 2 for appropriately pointing out our careless mistake; the relevant text has been deleted.

Reviewer 3 Report

Comments and Suggestions for Authors

I looked through the Daiji Nagayama manuscript with interest.

The topic of the study is relevant and the data obtained by the authors really adds new knowledge to the understanding of the mechanism of mutual connection between uric acid metabolism, the development of renal dysfunction and increased arterial stiffness.

  The research design is well planned, the data obtained are sufficient for correct reasoning and conclusions. Logical and understandable research methods were selected and a large representative sample was processed (n= 27,684). Statistics are correctly described and applied.

I agree with the authors' opinion that the associations of uric acid with KFD and systemic arteriosclerosis involve a complex causal relationship. Arterial stiffness, measured by the cardio-ankle vascular index, is a relevant marker of arterial stiffness and its association with the development of atherosclerosis. A three-pronged study of these risk factors is certainly relevant and necessary.

The results of our own research are quite competently compared with the literature on this topic, and logical conclusions are drawn. 54.5% of sources from the last decade, more than 32.4% from the last 5 years. However, 42% of sources are older than 10 years, which is a bit too much, but upon closer examination, the use of these sources is justified, since these are often basic publications on the topic.

Lines 148 to 162 were probably mistakenly left in the text of the manuscript; they need to be removed or explained why they are here, perhaps modified:

“The Materials and Methods should be described with sufficient details to allow others to replicate and build on the published results. Please note that the publication of yourmanuscript implicates that you must make all materials, data, computer code, and Diagnostics 2023, 13, x FOR PEER REVIEW 4 of 13 protocols associated with the publication available to readers. Please disclose at the submission stage any restrictions on the availability of materials or information. New methods and protocols should be described in detail while well-established methods can be briefly described and appropriately cited. Research manuscripts reporting large datasets that are deposited in a publicly available database should specify where the data have been deposited and provide the relevant accession numbers. If the accession numbers have not yet been obtained at the time of submission, please state that they will be provided during review. They must be provided prior to publication. Interventionary studies involving animals or humans, and other studies that require ethical approval, must list the authority that provided approval and the corresponding ethical approval code".

Minor note: in Table 2. Correlation of baseline CAVI with clinical variables.

In table line 2 LDL-C (mg/dL) an unreliable value is left - B - 0.000 r 0.965 the first source of literature is not written in full, there is no title of the article, only the authors and output data “1.Chen TK, Knicely DH, Grams ME. JAMA 2019;322(13):1294–1304. doi: 10.1001/jama.2019.14745.”

The design of the study is described in detail in the materials and methods, including measuring parameters of arterial stiffness and blood pressure, determining the relationship between serum uric acid and cardio-ankle vascular index.

Comments on the Quality of English Language

Moderate editing of English language required

Author Response

Reply to reviewer's comments.

We are grateful to reviewer 3 for the critical comment and useful suggestion that have helped us to improve our paper considerably. As indicated in the response that follow, we have taken the comments and suggestion into account in the revised version of our paper.

Comment 1.

Lines 148 to 162 were probably mistakenly left in the text of the manuscript; they need to be removed or explained why they are here, perhaps modified:

Response:

We thank reviewer 3 for appropriately pointing out our careless mistake; the relevant text has been deleted.

Comment 2.

“The Materials and Methods should be described with sufficient details to allow others to replicate and build on the published results. Please note that the publication of yourmanuscript implicates that you must make all materials, data, computer code, and Diagnostics 2023, 13, x FOR PEER REVIEW 4 of 13 protocols associated with the publication available to readers. Please disclose at the submission stage any restrictions on the availability of materials or information. New methods and protocols should be described in detail while well-established methods can be briefly described and appropriately cited. Research manuscripts reporting large datasets that are deposited in a publicly available database should specify where the data have been deposited and provide the relevant accession numbers. If the accession numbers have not yet been obtained at the time of submission, please state that they will be provided during review. They must be provided prior to publication. Interventionary studies involving animals or humans, and other studies that require ethical approval, must list the authority that provided approval and the corresponding ethical approval code".

Response:

We appreciate reviewer 3 for the valuable advice, and have once again confirmed the following points:

- The study clearly states that it is a retrospective cohort study.

- The ethics committee approval number and date.

- The description about the data availability.

Comment 3.

in Table 2. Correlation of baseline CAVI with clinical variables.

In table line 2 LDL-C (mg/dL) an unreliable value is left - B - 0.000 r 0.965

Response:

In response to the comment, the relevant part of Table 2, that is, the Standardized β for LDL-C in the Multivariate model, was revised from "0.000" to "< 0.001."

Comment 4.

the first source of literature is not written in full, there is no title of the article, only the authors and output data “1.Chen TK, Knicely DH, Grams ME. JAMA 2019;322(13):1294–1304. doi: 10.1001/jama.2019.14745.”

Response:

We apologize for the lack of title regarding the relevant literature. I appreciate reviewer 3 for pointing it out. The title was added.